# A Qualitative Study on Engaged Families’ Experiences with Long-Term Follow-Up Care in the Colorado/Wyoming Newborn Screening System

**DOI:** 10.3390/ijns10030061

**Published:** 2024-09-11

**Authors:** Stacey Quesada, Lauren Barringer, Marci K. Sontag, Yvonne Kellar-Guenther

**Affiliations:** 1Center for Public Health Innovation, Evergreen, CO 80439, USA; lauren.barringer@childrenscolorado.org (L.B.); marci.sontag@cphinnovation.org (M.K.S.); yvonne.kellar-guenther@cphinnovation.org (Y.K.-G.); 2Children’s Hospital Colorado, Aurora, CO 80045, USA

**Keywords:** newborn screening, long-term follow-up, genetic disorder, parents and caregivers, children with medical complexity

## Abstract

Understanding whether the long-term follow-up (LTFU) system is working for families is critical to measuring the success of newborn screening (NBS) and understanding why some families are lost to follow-up. Caregivers were recruited from six pediatric specialty care clinics. Data were gathered from caregivers via five focus groups and one individual interview (*n* = 24). Caregiver participants represented a wide range of children’s ages and conditions identified through NBS. While this is not the first study to gather caregivers’ input on LTFU, it provides a wide breadth of perspectives (e.g., metabolic, endocrine, hemoglobinopathy, etc.). When asked about goals for their children, caregivers identified health-related goals (i.e., children able to care for themselves, not hindered by diagnosis) and non-health related goals (i.e., defining themselves outside of disease, participating in sports, making friends). In describing the LTFU care they want and need for their child and the key factors that influence access and engagement, caregivers identified three themes: communication and relationships with providers; care team roles and factors; and care access and utilization factors. The themes identified are not disjointed; they are intertwined and illustrate the lived experiences of a sample of families engaged in LTFU care.

## 1. Introduction

Newborns born in the United States are universally offered newborn screening (NBS) for a wide range of conditions including hearing loss, congenital heart defects, endocrine disorders, hemoglobin disorders, inborn errors of metabolism, cystic fibrosis, spinal muscular atrophy (SMA), lysosomal storage disorders, and immunodeficiencies. Currently, conditions added to the Recommended Uniform Screening Panel (RUSP) undergo a rigorous review process, which includes a proven approach for treating the condition so that children identified through NBS have access to medical care to minimize the impact of the condition on their development and on their family’s quality of life [1,2,3,4]. One critical outcome of NBS is the connection to and receipt of long-term follow-up (LTFU) for families. 

LTFU has been defined as the period following confirmatory diagnosis after an abnormal newborn screen and through the remainder of the affected individual’s lifetime [5,6]. Since 1992, LTFU has been highlighted as a critical component to NBS [7]. Successful LTFU following an abnormal screen and confirmatory diagnosis plays an important role in optimizing the benefits of NBS while mitigating the harms. Several approaches to evaluating the long-term outcomes of NBS have been proposed, many of which leverage health record data to assess the child’s developmental and health outcomes longitudinally [5,8,9,10,11,12]. More comprehensive frameworks for assessing long-term NBS effectiveness, however, emphasize the importance of complementing clinical outcome measures with measures that capture family perspective and experience [13,14,15]. Including the family in determining what is appropriate LTFU is paramount. Families determine if they will obtain follow-up for their child, and they are most directly impacted by the LTFU their child receives.

Several studies have gathered input from families regarding their perspectives on the short-term follow-up period. These studies provide insight into the psychological impacts of false-positive or uncertain results, caregivers’ experience of being informed of a child’s abnormal newborn screen, knowledge of and attitudes towards the NBS process, and psychosocial implications of NBS results on families [16,17]. A handful of studies have derived recommendations for the LTFU system based on family input but many of these studies focused on one specific aspect of the LTFU period such as the quality of the primary care provider as a medical home or peer support among families living with similar conditions [18,19]. Other studies focused more directly on LTFU only sampled families living with a specific condition identified through NBS such as sickle cell disease or X-linked adrenoleukodystrophy [18,20].

Understanding whether the LTFU system is working for families is critical to measuring the long-term success of NBS and to understanding why some families are lost to follow-up [13,14,15,21,22]. The Health Resources and Services Administration states that children with special healthcare needs should “thrive in systems that support their families and their social, health, and emotional needs” [23]. Toward this end, this study aimed to learn what families’ goals are for their affected children, how they define successful LTFU, and what key factors influence their access to and engagement with LTFU as part of the Colorado/Wyoming newborn screening system. Five focus groups and one individual interview were conducted to capture input from a small sample of family caregivers. While this is not the first study to gather families’ input on their experiences with NBS, it provides a focus on experiences with and perspectives on LTFU among a cohort of families that represent the spectrum of NBS disorder categories (e.g., metabolic, endocrine, hemoglobinopathy, etc.) and a wide range of children’s ages.

## 2. Materials and Methods

### 2.1. Eligibility Criteria

Parents and caregivers of children who were diagnosed with a condition as a result of an abnormal newborn screen were eligible to participate. In addition, participants had to be between 18 and 89 years of age. Participants were recruited between March and June 2023 with the help of six specialty care clinics located at Children’s Hospital Colorado (Aurora, CO, USA), a large pediatric hospital. These clinics are contracted by the state NBS program to conduct follow-up with caregivers in their jurisdiction after an abnormal newborn screen.

### 2.2. Focus Groups

The focus group guide was developed through team conversations, consultation with an advisory committee comprised of specialty care providers and NBS program stakeholders, and a literature review. The guide included questions about what caregivers feel constitutes good LTFU care, barriers and facilitators to LTFU, what caregivers want changed with respect to their child’s LTFU care, and what goals caregivers have for their children. The guide can be accessed via Appendix A.

For recruitment, staff from six clinics responsible for NBS follow-up provided caregivers with flyers that included details about the focus groups. Flyers were distributed at clinic visits, through email, and through the online patient portal MyChart^®^. The flyers included a web address and QR code directing interested participants to an online interest form.

Five semi-structured focus groups (*n* = 23) and one individual interview (*n* = 1), lasting approximately 90 minutes each, were conducted with caregivers online using the Zoom videoconferencing platform. Focus groups ranged in size from three to six participants. Caregivers were invited to a given focus group based on their availability, not based on their child’s age or condition thus leading to heterogenous groups. All focus group participants received a USD 150 gift card for their time. In all focus groups, there was one facilitator and one note taker. A certified translator provided simultaneous translation for a participant whose preferred language was not English. Focus group recordings were transcribed by a transcription service. Transcriptions were cross referenced with the notes to ensure accuracy and completeness. At the end of each focus group, participants were asked to complete a short survey to capture demographic information which included questions about their child’s disorder, age of child living with a genetic condition, total number of children, number of children with complex and/or chronic medical issues, whether the family’s needs are being met with respect to their child’s medical care, the average time to commute to their child’s medical providers, and whether they are able to attend all of the recommended well-care and specialty care visits with their child’s medical providers.

### 2.3. Analysis

All focus group transcripts were coded and analyzed using NVivo Version 14.23.2 (46). An initial codebook was developed based on a literature review and a round of cursory coding performed by all three coders on one of the focus group transcripts. The codebook was continually revised and expanded during each round of coding. Inter-rater reliability was ensured by having one focus group coded by all three coders and four focus groups coded by two coders. Coding pairs reconciled all codes and resolved any discrepancies. The coding team met upon completion of coding to review the final codebook and consolidate codes where appropriate.

The study team coded the transcripts using content analysis [24]. Prominent themes were identified if at least 30% of the participants (*n* ≥ 7) reported the same concept. Quotes supporting the findings were identified and have been included with the results.

## 3. Results

### 3.1. Participants

Forty-seven caregivers expressed interest in participating in a focus group; twenty-four could be reached by the study team and were scheduled for a focus group. Due to a scheduling conflict, one of these caregivers ended up participating in an individual interview; the other twenty-three caregivers each participated in one focus group. Table 1 summarizes the focus group and interview participants’ characteristics. The size and composition of each focus group and interview are detailed in Table 2. It should be noted that the caregivers who participated in this study are a sample that is engaged with LTFU care, owing in large part to the recruitment strategy which advertised the focus groups at the specialty care clinics where families receive LTFU care.

### 3.2. Caregivers’ Goals for Their Children

When asked what their goals are for their children, participants most often identified health-related goals (mentioned 34 times by 18 participants). These goals included their child being able to care for themselves in the future (i.e., their child having more autonomy in their care, being able to advocate for themselves, being able to access and utilize resources to help them with their condition, reduction in medications needed, not having to deal with insurance issues) and their children not being hindered by their diagnosis, including their condition being eradicated or cured, living a long life, and meeting developmental goals.


*“… I would like her to be independent in her own self-care … I think all kids go through a rebellious, falling off the wagon, and ‘I’m not doing that anymore; I’m done with that.’ And luckily, we’re on a drug that hopefully will eradicate the need to [do medical procedures]. But we’re still doing those things. And I mean, if that continues to have to be a need, I want her to take ownership and feel like empowered instead of just dragged down by it … I’d like to see her just have a healthy relationship with it. And just think of it in a way of, ‘oh, how great I have these things to help me be healthy’ instead of like, ‘oh, what a drag. I have to do this.’”—Study participant*


Several non-health related goals (mentioned 22 times by 16 participants) were also expressed by caregivers, including goals specific to persons living with a condition such as the child being able to define themselves outside of their disease, being able to educate others on their disease, and communicating using a tablet. In addition, caregivers communicated goals for their children not tied to living with a genetic condition, such as doing typical kid things (i.e., learning to drive, making friends, participating in sports, attending sleepovers), feeling included in life, having good relationships with people outside the immediate family, being a nice person, living a normal life, and becoming more independent.


*“I think advocating for themselves, taking care of their bodies, preventing as much illness as they can, listening to their bodies as well. And we teach our boys you are not your disease. The disease is something that you live with. And so, my hope for them is just for them to live full long, happy lives. My son is in competitive sports, and I would hope that he is able to continue to do that as long as it finds him happiness. And I don’t want his physical ailments to prevent him from doing the things that he wants to do with his life.”—Study participant*


### 3.3. Key Factors That Comprise or Influence Long-Term Follow-Up Care

In discussing what comprises successful LTFU care for their family and the factors that impact LTFU access and engagement, family caregivers identified factors that can be summarized into three themes: communication and relationships with providers; care team roles and factors; and care access and utilization factors. Communication and relationships with providers, and care team roles and factors, were the most prominent themes among participants, both mentioned at least once by each participant (113 and 100 mentions by participants, respectively). The theme of care access and utilization factors was only slightly less prominent, discussed by 22 out of the 24 study participants (85 total mentions).

Based on the input from caregivers in this study, a model was developed to depict the interplay of the themes identified in accessing and engaging with LTFU care from the caregiver’s perspective (Figure 1). In the model, the caregivers play a vital role, with their family at the core of LTFU. Within the family unit are the parents’/caregivers’ goals for their children which may guide the family and the rest of the care team in caring for the child as a whole person who is not defined solely by their disease. The family and provider relationship, which is built through communication, is also at the core of the model. Communication and relationships between the family and providers, along with supportive factors of psychosocial support for the family and knowledge and education, comprise LTFU care in the model. LTFU care operates inside an environment that is impacted by access and utilization facilitators and barriers (i.e., scheduling, insurance, elderly parents, other children, work, and transportation to the clinic).

#### 3.3.1. Communication and Relationships with Providers

##### Communication with Providers

All participants identified communication as a key component to LTFU, which is why it is central to the model. Caregivers collectively highlighted a triad of communication as an essential piece of LTFU care. The triad consists of communication between the family and the specialty provider, the specialty provider and other care providers (e.g., primary care, emergency care, hospitalists, etc.), and the family and other care providers. According to caregivers, communication needs to be bi-directional, flexible, respectful, timely, and must take place between each of the nodes in the triad.

Caregivers discussed the importance of being able to reach out to the specialty clinic with questions or concerns, whether via a patient portal, email, phone, or in-person, and the specialty clinic communicating with the family, including responding quickly to questions, going over a timeline of what to expect and what will need to be carried out next, and asking the family for feedback. In addition, caregivers expressed a desire for providers on the child’s care team to communicate with each other about the child’s care and for other providers (i.e., non-specialty providers) to communicate with families in a way that respects their expertise with their child’s condition and intimate knowledge of their child’s health. This communication triad can be seen in the context of the broader LTFU model in Figure 1 and in more detail in Figure 2, which includes the specific factors that contribute to its success, as noted by caregivers.


*“Quality care for me has been the response time to questions, because we have to feed them every day and sometimes, especially in the beginning, in your first year of PKU, the worry that goes along with what, if what you’re doing is going to be damaging, I would say quick response time is really important.”—Study participant*



*“And before I leave [the appointment], they go over when’s their next appointment. They also go over their school plan and things like that. So as the kids have gotten older … it’s new things to consider. So, daycare versus school, now it’s sports, but I think it’s the scheduling the appointments and scheduling them in advance and then knowing how frequently we need to go in. And each appointment may be blood work at this appointment, another appointment may be a doppler, so I know when to expect those every couple months or per year.”—Study participant*


Caregivers identified communication breakdowns in the triad as barriers to LTFU. These breakdowns included caregivers not feeling listened to by those performing medical procedures, families not having a way to communicate with a provider directly, providers not communicating with others on the care team, and inconsistencies in the specialty clinic’s communication with the family. Many caregivers had negative experiences with the phlebotomists or nurses performing their child’s blood draws; they felt their preferences for the procedure were ignored, which led to discomfort and anxiety for their child. Given the frequency of blood draws required for these children, this significantly impacted their LTFU experience.


*“One time we went to the emergency room, [child] was really sick, and I told the nurse this arm is really the best one to do [the blood draw]. [Nurse] didn’t listen to me, and she did it on the right and the arm was really swollen … Now, [child] gets really anxious about just going to the hospital because of drawing her blood.”—Study participant*



*“Our biggest issue or my biggest issue is when we go to get the blood draws. Phlebotomists don’t want to listen to me, which is kind of annoying because I did that for a lot of years … And it’s a constant fight every single time of telling them, just use her other arm.” —Study participant*


##### Relationship between the Family and Providers

Focus group participants identified the relationship between the family and their children’s providers as a key factor in the quality of LTFU care. When describing good relationships with their providers, caregivers used the terms “respectful” and “compassionate” and noted an appreciation for providers taking the time to get to know their child and the family. Positive family-provider relationships contributed to family empowerment in LTFU decision-making, giving caregivers the confidence to push back on providers when they felt it was necessary.


*“I feel like our doctor is really cool. I have a good rapport with her. And they like to say, ‘Oh, if your child’s had a cough for five days, let us know, we’ll put them on antibiotics.’ I’m like, ‘No, I’m not doing that.’ … So, I kind of feel it out … And I stay in touch with them. And [doctor] is very respectful of the decisions I make.”—Study participant*


Conversely, several caregivers described negative aspects of their relationships with their child’s providers including not feeling heard, not feeling their family choices or values were supported, or feeling judged by their providers. In addition, multiple caregivers reported feeling uncomfortable disagreeing with their child’s providers, one even describing it as being “scared of pushing back.”


*“I felt it two different ways and the pressures in both ways about what you’re doing and the lack of understanding from [the providers] is still there. And I wish that that would change and give more support for whatever choice a family makes one way or the other. And I felt no matter what, there was judgment and with a child that has additional medical needs, that pressure is even greater and even harder, and you feel even more alone in trying to make the right decisions for your child.”—Study participant*


#### 3.3.2. Care Team Roles and Factors

##### Caregiver’s Role and Needs

Study participants expressed that they play a central role in accessing and maintaining LTFU for their child. Some of the tasks associated with this role include advocating for their child, staying on top of appointments, communicating with the care team including challenging the doctor when needed and seeking input from other providers, building relationships with their child’s providers, and educating non-specialty providers on their child’s condition and needs.


*“I’ll just ask to get a blood draw done, even if [provider] tells me, ‘No, that’s not a symptom.’ And there was one time that [provider] was off, and [child] did need a dosage change, but other than that I try to listen. I try not to be that overbearing parent, but I am also the voice for my daughter. So, I’m like, ‘No, let’s just go get [child] checked.’”—Study participant*


Lack of knowledge among non-specialty providers was cited by several participants as something that makes it difficult to obtain the care they need for their child. Many caregivers reported frustrations with hospital or primary care providers who do not understand their child’s disorder and are therefore not equipped to treat them appropriately. In these circumstances, parents and caregivers stated that they would have to advocate for their child by educating providers on their child’s condition.


*“The only times where we’ve had disagreements have been with our pediatrician and nurses and things like that. And it has gotten I wouldn’t say combative, but definitely from both sides where we’re voicing our opinions and the pediatrician or the nurses in the hospital think that they know what they’re doing and then we call them out, ‘Actually, that’s not the cutoff for those labs,’ or things like that. It goes back and forth and then they’ll bring in the doctor or they’ll bring in the residents and then they’ll try to say the same thing. And we’re like, ‘No, that’s not the cutoff.’”—Study participant*



*“They [non-specialty providers] have no idea what it is, so it’s like you have to explain it to them.”—Study participant*


While the need for educating non-specialty providers was often discussed in the context of hospital or primary care, caregivers were also confronted with a lack of knowledge when calling to schedule their child’s appointments. Schedulers who did not understand the nuances of LTFU protocols, and did not know how to contact specific schedulers responsible for handling LTFU appointments, would give parents an extremely inaccurate estimate of when their child could be seen.


*“We’re hitting this barrier where I had called, and the scheduler just had no clue, and she didn’t have access to the right schedule so she’s telling me that the doctor doesn’t have an appointment for a year and a half and I’m like, that’s ridiculous.”—Study participant*


Caregiver burnout, denial, fear, feeling overwhelmed, or previous negative medical experiences were identified by caregivers as barriers to staying engaged in LTFU care.


*“And when you find out that your kid has a genetic condition, it’s overwhelming and then you don’t feel like … the doctors … support your family values. And at some point, I think some parents just give up because it is so much to manage mentally, especially if you have more than one kid or even if you just have one kid that you’ll have seventeen specialists in addition to the normal doctors. It is just so overwhelming that you just withdraw … because you have no other option.”—Study participant*


Caregivers noted that connection to other families living with a similar condition (e.g., support groups) and psychosocial supports such as access to a social worker and psychological support for the child and the family were supportive factors for families accessing and maintaining LTFU.


*“It is very isolating. But what was great was that I got connected really early on to a Facebook page that was just directly for children with my son’s disease … And then also in the area, the specialist had given me names to people that were close enough that had the same disorder as my son. So, I thought that was really helpful and comforting too to just know that there are people that have the same issues or feeding issues.”—Study participant*



*“Or at least even to have a support group with people even that have children with the same condition, that would be helpful, too. Maybe just so that I’m not bothering the doctor every single chance I get to, ‘Hey, this is just a developmental thing,’ or, ‘Hey, this is an actual what she’s got type of thing?’”—Study participant*



*“…the emotional impact of that transition into rare disease life or special medical care, I think that’s something too, that getting people the support that they need to be able to effectively care for their children. It’s more than just the technical knowledge. I think it’s making sure that they’re in a good place to be able to do that.”—Study participant*


Caregivers also reported finding comfort in being educated on the disease and what to expect at each stage. In addition, caregivers discussed how their knowledge of the disorder allowed them to better advocate for their child and to openly disagree with providers, especially in circumstances where caregivers sensed a provider’s lack of knowledge.


*“To me, it’s giving the parents the tools that they need, the knowledge and tools that they need to be able to manage that independently.”—Study participant*



*“I’ve disagreed with hospital staff, not specialists. And I think probably with all of us, we’re taught to really advocate for our children in places where their specialists aren’t there, and nursing staff in particular who are in the room more. And I’ve had to really advocate. I’m on it in a hospital situation especially a lot of times we’re not in a children’s hospital, we are up in a non-children’s facility and so I certainly just rely on my knowledge and the disorder as I know it.”—Study participant*


##### Team of Providers Treats the Whole Child

According to focus group participants, LTFU care, although led by a specialist, is delivered by a team of providers, medical and non-medical, who treat the whole child. A wide range of providers including specialists, primary care providers, dieticians, physical therapists, pharmacists, condition specific foundations, mental health professionals, social workers, and genetic counselors were highlighted by caregivers as being part of their child’s care team and contributing to their LTFU.


*“So, they connected us with the CF Foundation quickly and made inroads there. Medications are extremely expensive for people with CF, so insurance support, pharmacy support, and then just the psychology, the emotional support.”—Study participant*



*“It is a whole team of people that we see on the regular that we can request to see more often. Actually, they’re pretty cool because they do treat the whole person. And it’s like right now I’m like, we don’t need a psychologist so much because she’s a little kid. But man, I understand why that’s such an important part of the team. As she gets older and feels isolated with her disease as a teen or something, that’s a thing. And so, they have so many people in place to manage every aspect of the disease, emotional, physical. That’s huge.”—Study participant*


With a large team of providers, however, there are more opportunities for turnover in care team members, which can lead to gaps in families receiving the care they need. Many caregivers expressed frustrations with the turnover they experienced among the various providers in their child’s LTFU care team.


*“And we had an amazing mid-level that we worked with, and [provider] was just on point, she was on top of the research and when we would reach out to her, she could fix things, she could answer questions, and then she left … And we’ve had a lot of struggles since that has happened, a lot of struggles trying to get [child’s] prescriptions refilled.”—Study participant*


A role often missing from their child’s care team, as noted by several study participants, is that of a care coordinator. Caregivers expressed the desire for a care coordinator to help manage their child’s care across the team of providers and the high volume of appointments. Without care coordinators, caregivers assume this role, which can be time consuming, overwhelming, and detrimental to their other commitments such as work and caring for other family members.


*“So to have some of that, ‘Hey, here’s things to be checking,’ even if it’s just a checklist to say, ‘Get the labs done, get this scan done, get the test done, get the evaluation scheduled,’ to have that, even if it’s not a worry, you’ve at least done the example and to have that confidence that somebody … is reviewing your kid’s medical record from a global top-down perspective.”—Study participant*


#### 3.3.3. Care Access and Utilization Factors

Participants identified a few key factors that influence whether a family can access or utilize the LTFU care that is available. These include insurance and logistical factors that impact caregivers’ ability to schedule and/or attend LTFU care appointments.

##### Insurance

Insurance was identified by study participants as a barrier to LTFU or as something that could be improved in their LTFU experience. Caregivers discussed how lack of insurance, poor insurance coverage, complex enrollment processes, and difficulties obtaining treatment authorization make it difficult for families to obtain the care they need. Specifically, many of the caregivers in the study experienced problems obtaining insurance approval for their child’s medications or medical food, even after it had been approved previously.


*“… my kiddos, also, they take the same medications every single month that they’ve been taking since they were born and at the pharmacy every single month there’s an issue with the insurance. And we have private insurance, and they always have to call it in, and it always takes days for them to get it figured out.”—Study participant*


Once they completed the complex enrollment process, public insurance programs, such as Medicaid, Children’s Health Insurance Program (CHIP), and TriCare were considered very beneficial to families’ LTFU care.


*“The CHIP program we went through when she was little, getting evaluated … I didn’t understand everything I was doing. I felt like I had to go through a lot of hoops … but it’s part of their program, and it’s kind of putting out a safety net, I feel like. When they get someone into certain programs, they really do investigate all the needs and try to support the families in that way. And I mean, I think that’s pretty spectacular.”—Study participant*



*“Being established with Medicaid has been pretty sweet.”—Study participant*


Given the complexities of enrolling in public insurance or obtaining authorization for critical procedures and medications, caregivers see value in having access to insurance specialists who can help families navigate insurance issues.


*“… to have maybe an insurance specialist or something that could navigate all the different insurances and help families get on Medicaid and help navigate Medicaid and if there’s other funding sources out there. Because, again [with] PKU … it’s just an expensive disorder and [the] clinic, while they try and offer suggestions, I think they’re so busy dealing with just the medical aspect of it. It might be nice to just have somebody that can help with the insurance and financial.”—Study participant*


##### Logistical Factors

Caregivers also identified logistical factors that are barriers to accessing and maintaining LTFU. The most frequently mentioned logistical issue was difficulty attending their child’s medical visits because of geographic distance, weather, issues with transportation, or no internet access for telemedicine visits. The family’s obligations to things outside their LTFU world such as caring for other children or elderly parents, caregivers’ jobs, and their children attending school were also barriers.


*“Yeah, it’s really hard. Actually, my husband lost his job because … of the whole hospital stay … My son has the medical condition, has siblings, so [caregivers] have to share the duties.”—Study participant*



*“I spent hours upon hours upon hours waiting in waiting rooms and I was also a full-time working parent and so it became almost impossible. I had to quit my job because of the amount of needed appointments for two different issues that were going on along with the care that was needed.”—Study participant*


Scheduling difficulties pose another layer of logistical complexity. Caregivers discussed several issues with scheduling including not being able to get in touch with the right scheduler, not being able to bulk schedule appointments on the same day, difficulties rescheduling appointments, and long wait times for appointments.

Caregivers expressed a desire for more flexibility with frequency and format (i.e., virtual, in-person) of clinic visits to offset the difficulties posed by the logistical factors mentioned above.


*“… I think spreading out those appointments would be helpful and just going to the lab, but then just doing a virtual visit for the results.”—Study participant*



*“I mean, you can’t see PKU. It’s in their blood, it’s in their body, so he doesn’t really need to be physically seen. So, the telehealth is nice, because it cuts down on drive time.”—Study participant*


## 4. Discussion

This study provided insight into the goals caregivers have for their children who were diagnosed with a disorder resulting from an abnormal newborn screen, what caregivers want and need with respect to their child’s LTFU care, and the factors that influence families’ access to and engagement with LTFU care. The caregivers who participated in this study represent a wide range of disorders currently identified through newborn screening and ages of affected children.

The caregivers’ goals for their children included both health and non-health related goals, highlighting caregivers’ desire for their child to be viewed as a whole person when receiving LTFU care for a potentially life-altering diagnosis. These parents wanted their children to be able to care for and advocate for themselves, a role that parents have now, but they also wanted their children to be able to do typical childhood things like participate in sports, attend sleepovers, and learn to drive. With respect to their child’s LTFU care, family caregivers identified three overarching themes: communication and relationships with providers; care team roles and factors; and care access and utilization factors. These themes are supported by existing literature.

The most prominent theme in this study was the importance of the relationships and need for an effective triad of communication between the family, the specialist, and the rest of the child’s care team. These relationships and the accompanying communication triad are an essential component of equitable and sustainable LTFU. There is a strong association between best practices in physician–family communication and positive patient outcomes [25]. Providers applying best practices in patient-centered communication as part of LTFU build positive family-provider relationships, boost trust between families and providers, and increase caregiver’s knowledge of their child’s disorder and their capacity to advocate and care for their child [26].

Communication between all care providers is also crucial. Caregivers in this study described the diverse team of providers who treat their child, not unlike the multidisciplinary teams treating the broader population of children with medical complexity (CMC). These teams include physicians, advanced practiced providers, nurses, social workers, therapists, dieticians, support staff, and community providers, all of whom play a vital role in the treatment of the whole child yet simultaneously introduce more opportunities for inconsistency, complexity, and turnover in the child’s care [27,28,29]. In the absence of direct communication between providers, it is often up to the caregiver to relay important medical information. While caregivers in this study viewed advocacy and educating non-specialty care providers as important tasks, this places the onus of medical literacy on the caregiver who, given their life’s stressors and circumstances, may or may not have the “absorptive capacity” to process and relay information accurately [17,18]. Ideally, LTFU care practices will have designated care coordinators who can serve as the family’s single point of entry to the complicated, ever-changing care system, allowing caregivers to focus first and foremost on being a caregiver for their child [17,30].

The caregiver’s role in LTFU is time consuming, overwhelming, and can be plagued by fear, denial, and burnout. They must juggle the demands of being a parent, a skilled care provider, and a care coordinator for their child in a complex multidisciplinary system, which can make it difficult to manage other logistical constraints such as caring for other family members and maintaining employment [31,32,33]. The LTFU system can provide psychosocial supports such as peer support groups and psychological assistance to support families as they manage and sustain LTFU care [19,20,26,31].

Another key factor for the care team is education on the child’s disorder. Education for caregivers enables them to manage their child’s condition and ensures they have the capacity to properly advocate for their child, which are important skills for their role on the care team [26,33,34]. Providing education on disorders currently identified through NBS to all care providers on the team has the potential to improve families’ experience with LTFU and can reduce the burden on caregivers [17,19,20,26]. Establishing foundational knowledge of these disorders among families and ancillary care providers allows them to speak the same language when it comes to the child’s care, thus fostering trust and respect and equipping the whole care team with the resources and knowledge to maximize the benefit and minimize the harm experienced by families and children accessing and receiving LTFU care.

The caregiver’s relationship and communication with providers and the education of the care team on the child’s condition are moot, however, if families cannot access or utilize LTFU care. For families living with disorders identified through NBS, lack of insurance, inadequate insurance coverage, complicated public insurance enrollment, and difficulties with insurance authorizations are significant barriers to LTFU care that can involve numerous specialty clinic visits, expensive interventions, and frequent tests [35,36,37,38,39]. Lack of insurance has been found to be associated with higher odds of being lost to follow-up [30,35,36,37,38,39]. Insurance specialists or coordinators can help caregivers navigate the complex world of insurance coverage and enrollment, thereby reducing the risk of not being able to access the care they need for their child. For clinics that do not have access to insurance specialists, it may be beneficial to educate an existing staff member on frequently encountered insurance issues and the corresponding solutions or resources that may help families overcome these obstacles.

### Limitations

There were several limitations to this study. Based on recruitment strategies, the study only reached families who are currently engaged with the Children’s Hospital Colorado healthcare system and/or utilize e-mail or the online patient portal. As a result, this study cannot speak to the factors that contribute to those who are lost to follow-up. In addition, most participants were English-speaking caregivers. The study included only a very small sample of caregivers. As a result, the study cannot speak to variations that may or may not exist between different conditions. Future research should be conducted with a larger and more diverse sample of family caregivers to be able to generalize results to the broader population of family caregivers in the United States.

## 5. Conclusions

Caregivers identified three themes in defining the LTFU care they want and need for their child and the key factors that influence access and engagement: communication and relationships with providers, care team roles and factors, and care access and utilization factors. These themes are not disjointed; they are intertwined, interdependent, and illustrate the lived experiences of a cohort of families engaged in LTFU care in the Colorado/Wyoming newborn screening system. Embedded within these themes are the caregivers’ health and non-health related goals for their children. Caregivers are responsible for advocating for their child, scheduling appointments, communicating with providers, transporting their child to and from clinic visits, monitoring their child’s health on a day-to-day basis, and managing health insurance. An evaluation of the effectiveness of NBS must consider the complicated and protracted experiences of the families receiving LTFU. Furthermore, in striving to maximize the benefits, it is imperative for the NBS LTFU system to provide resources and support that ensure families’ access to the long-term follow-up care they want and need for their child.

## Figures and Tables

**Figure 1 IJNS-10-00061-f001:**
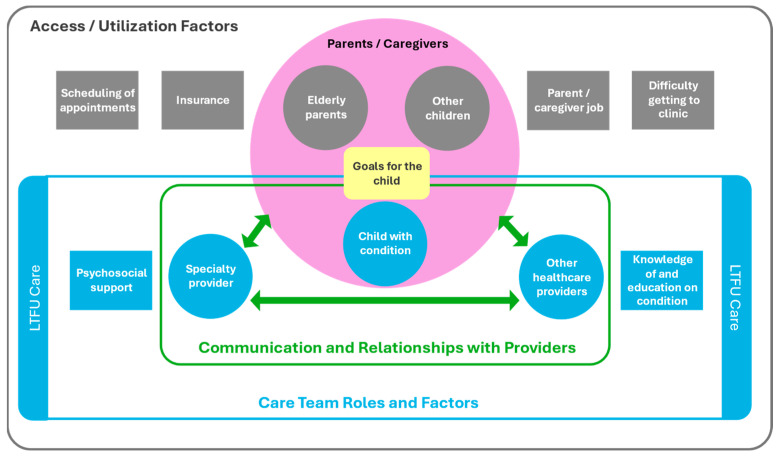
A model of LTFU access and engagement, from the caregiver’s perspective.

**Figure 2 IJNS-10-00061-f002:**
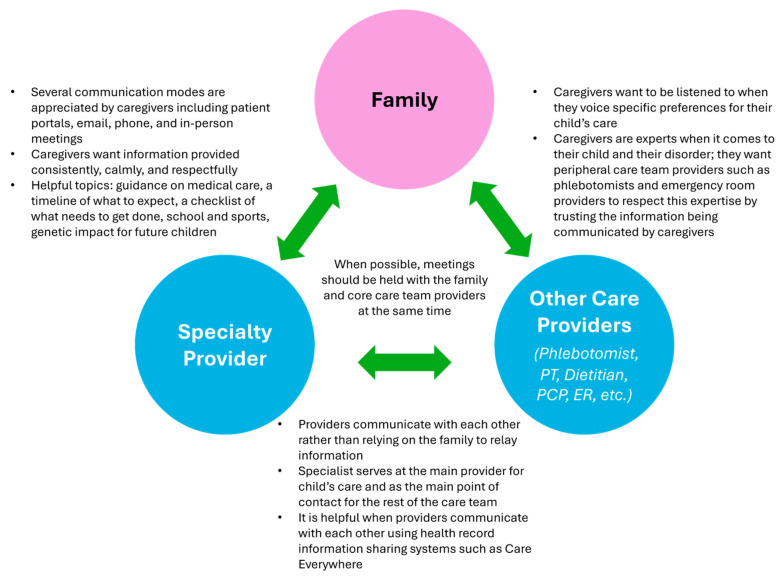
The LTFU communication triad, as described by caregivers.

**Table 1 IJNS-10-00061-t001:** Participant characteristics.

Participant Characteristics		*n* (%)
Category of Genetic Condition with Which Child is Diagnosed	Metabolic	13 (54.2%)
	Cystic fibrosis	4 (16.6%)
	Hemoglobinopathy	4 (16.6%)
	Endocrine, SMA, or SCID	3 (12.5%)
Primary Language	English	23 (95.8%)
	Spanish	1 (4.2%)
Age of Child at Time of Data Collection *	Birth to 3 years	2 (8.3%)
	3 to 5 years	7 (29.2%)
	5 to 10 years	9 (37.5%)
	10 to 15 years	2 (8.3%)
	Unknown	4 (16.6%)
Number of Children Currently	1 child	3 (12.5%)
	2 or more children	19 (79.2%)
	Unknown	1 (4.2%)
Number of Children with Complex and/or Chronic Medical Issues (CCMI)	0 children with CCMI **	2 (8.3%)
	1 child with CCMI	18 (75.0%)
	2 or more children with CCMI	4 (16.6%)

* The child who was diagnosed with a disorder as a result of an abnormal newborn screen. ** It is assumed these parents/caregivers do not feel their child’s condition is complex or chronic; their child’s condition may be well managed at this point.

**Table 2 IJNS-10-00061-t002:** Size and composition of the focus groups and interview.

Focus Group/Interview Identifier	Category(ies) of Disorders Represented by Participants	Number of Participants
1	Hemoglobinopathy Endocrine, SMA, or SCID	3 1
2	Cystic fibrosis	1
3	Hemoglobinopathy Endocrine, SMA, or SCID Cystic fibrosis	1 1 1
4	Endocrine, SMA, or SCID Cystic fibrosis Metabolic	1 1 3
5	Cystic fibrosis Metabolic	1 4
6	Metabolic	6

## Data Availability

Data are unavailable due to privacy restrictions. This report includes quotes to support the major findings.

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
