# Peer review of "A Qualitative Study on Engaged Families’ Experiences with Long-Term Follow-Up Care in the Colorado/Wyoming Newborn Screening System"

_2409-515X, 2024, doi:10.3390/ijns10030061_

Round 1

Reviewer 1 Report

Comments and Suggestions for Authors

This manuscript reports a study of caregivers of children in LTFU for conditions identified through newborn screening. The total number of participants was relatively small but the diversity of conditions represented is good.  The methods used are appropriate, the topic is important, and the manuscript is well-written.

A challenge for this type of report is to avoid descriptions that come across as  a potpourri of responses or a catalogue of frustrations.  I think the authors have avoided this with their synthesis into several themes and their use of quotations.  However, the authors have not included any quotations or findings that are more sharply critical.  The findings are primarily phrased in aspirational and suggestive terms. Are there any findings about bad or inappropriate practices that providers must avoid?  Might the findings be more impactful if the phrasing was more like, "Caregivers consistently appreciated X but many had strongly negative views of Y."  Some of the broader issues are largely beyond the control of providers, such as insurance coverage, so it would be ideal if the manuscript was written to offer as much specific guidance to providers as possible on issues that they can control.  This is not to suggest an extensive re-write, only a consideration of tone and examples in a few places, if supported by the data.

Reviewer 2 Report

Comments and Suggestions for Authors

I found the manuscript "Families’ Views on Accessing and Utilizing Appropriate Newborn Screening Long-Term Follow-up" to be well written but to be lacking in numbers of participants. The 24 focus group participants seem to have been the primary focus given the large number of focus group examples versus the survey respondents (something like 28 vs. 2). Additionally, based on the table, most patients were not diagnosed before 3 years of age, which is not characteristic of most newborn screening programs.  In addition to requesting more information on this point, there are many other points that need to be addressed before the paper is suitable for publication. While I agree with the findings generally, clarification in the text about the issues raised should help me (and others) to better understand these results.

Line 27 – The term ‘infant’ is often used to refer to babies a bit older than newborns. For consistency, please use the term ‘newborn’ here. Suggestion: “Newborns in the United States are universally offered screening shortly after birth for a 27 wide range of conditions …”

Lines 30-32 – It is not true that most of the conditions on the RUSP have undergone a ‘rigorous review process’ since most were included in the original ACMG recommendations that did not include the rigor of the current process (evidence review) and only looked at items like inclusion in a multi-analyte screening process, etc. Suggestion: “Currently, conditions included on the Recommended Uniform Screening Panel (RUSP) undergo a rigorous review process, which includes a proven approach for treating the condition …”  References: [1] Calonge N, Green NS, Rinaldo P, Lloyd-Puryear M, Dougherty D, Boyle C, Watson M, Trotter T, Terry SF, Howell RR; Advisory Committee on Heritable Disorders in Newborns and Children. Committee report: Method for evaluating conditions nominated for population-based screening of newborns and children. Genet Med. 2010 Mar;12(3):153-9. doi: 10.1097/GIM.0b013e3181d2af04. [2] Kemper AR, Green NS, Calonge N, Lam WK, Comeau AM, Goldenberg AJ, Ojodu J, Prosser LA, Tanksley S, Bocchini JA Jr. Decision-making process for conditions nominated to the recommended uniform screening panel: statement of the US Department of Health and Human Services Secretary's Advisory Committee on Heritable Disorders in Newborns and Children. Genet Med. 2014 Feb;16(2):183-7. doi: 10.1038/gim.2013.98.

Lines 36-38 – Both references listed define long term follow-up as lasting throughout the lifetime of the individual and do not limit it to school age or 18. Please correct.

Lines 38-39 – The importance of long term follow-up in newborn screening has been argued since at least 1992. Please see following reference and correct: See pp. 143 and 145 of Therrell BL Jr, Panny SR, Davidson A, Eckman J, Hannon WH, Henson MA, Hillard M, Kling S, Levy HL, Meaney FJ, et al. U.S. Newborn screening system guidelines: statement of the Council of Regional Networks for Genetic Services.  Screening 1992;1:135-147. [Google Scholar]

Line 58 – Please clarify reference to sickle cell by adding the term ‘anemia’ if you are rereferring to Hb SS disease or ‘disease’ if you are referring generally to one of the sickle cell diseases or to the grouping.

Line 62 – Please add reference 5 to the group of references and consider the addition of the following reference: Hoff T, et al. Long-term follow-up data collection and use in state newborn screening programs. Arch Pediatr Adolesc Med. 2007 Oct;161(10):994-1000. doi: 10.1001/archpedi.161.10.994.

Lines 64-70 – There is no mention of the use of focus groups or surveys to meet the study goals or a clear statement of what the final goals are. Please include some mention of both of these items. Also, it would be nice to include the surveys and the focus group guide (or discussion topics) as supplemental material. The number of focus group participants is very small and your later discussions seem to have relied almost entirely on these few caregivers for your results.  Please introduce this approach so that the reader Is not left wondering about the study’s validity based on so few participants.

Line 79 – Please add the n value for participants in each group as in the abstract. Please add a sentence describing the number of patients with each of the four categories of condition for both the focus groups and the survey respondents.

Line 87 – Please describe the size of the focus groups, for example, “Six semi-structured focus groups, lasting approximately 90 minutes and varying in size from 7-24 participants, …..” Also can you differentiate between the subjects for the different focus groups or were they all the same?

Lines 133-144 – The information given under Participants, including Table 1, is really information that should be added to the information on Focus Groups under Materials and Methods. Please clarify whether Sickle Cell means Sickle Cell Anemia or the broader category of Sickle Cell Disease. Table 1 should be discussed more fully in Results – Is it really the case that only 2 of the cases in the focus groups and 20 survey cases were diagnosed before age 3? This needs fuller discussion since age of diagnosis can play a significant role in follow-up and follow-up expectations. The impact of age at diagnosis may also need discussion in reviewing expectations, etc. Newborn screening is designed for early diagnosis and treatment and this group of patients appears to be outside of the norm.

Line 139 – Please expand to include information on how many focus groups were attended by each. As written, it appears that the focus groups each had about 4 attendees (6 x 4 = 24).  Is this correct? Please clarify.  For example, “The 24 focus group participants attended at least one group meeting, x attended two, y attended 3, z attended 4 …”  or something similar.

Line 142 – Please either give the range of response rates for the 6 clinics or change the wording to say that the total response was 14%.

Lines 172-185 – I assume these comments are meant to emphasize what was already said.  If so, a short (less than a full sentence introduction should be inserted to explain this to the reader. While of some interest, I did not find these examples to add much to what was already clearly stated. With these and other example comments, it would help to know the condition of the child in addition to the fact that they were a focus group participant.

Lines 215-217 – Can you please comment either here or in the discussion on why only 1 survey respondent noted communication as a key component of follow-up while all focus group participants mentioned it? Why do you consider it central when 129 of 130 survey participants did not? It seems that this and most of the rest of the Results were from the focus group discussions and the survey results were not significant given that all examples seem to be from the focus groups (28 focus group examples and 2 survey examples, by my count) and you have noted at the end that the quotes are given to support the major findings since data are not available. There needs to be some discussion about why data are not available, and why the quotes support the major findings (of the focus group discussions?).

Lines 572-579 – Please include the fact that the study included only a very small sample of caregivers. The numbers are too small to say whether there are variations between different conditions, or were you able to tease this out?

Reviewer 3 Report

Comments and Suggestions for Authors

The authors’ chosen topic of newborn screening long-term follow-up is of importance and the selected family caregivers’ quotes enhance the policy imperative. From the perspective of this reviewer, the body of the manuscript and thematic analysis are strong overall as is the methodology employed. In contrast, from the perspective of this reviewer, there are a few significant areas that need to be considered and addressed, as noted below.

This reviewer found the title problematic on several points, further reflected elsewhere: 

·      The term “appropriate”  as used here combined with LTFU. This can be perceived as overly value-laden and is poorly defined in the context of “appropriate long-term follow-up” (appears in the title and within text body) given there is arguably little consensus of this formula. Whereas there is somewhat more general agreement regarding “appropriate care” models as used in another recent IJNS article on LTFU, when these same authors used “Appropriate “ in combination with Care.

·      “Families’ Views” combined with “on accessing and utilizing” per the title: appears to this reviewer overly broad and too inclusive, given these only involved a cohort of families who had followed through with LTFU. An import cohort is not ptrdrnt and this should be clear,For similar reason, the subgroup should be made very clear much earlier than when noted in limitations . Moreover, the reference point should be“some” families in the text body rather than suggest  “collectively illustrate the lived experiences of families at the epicenter of LTFU care.”

·      From this reviewer’s perspective, this manuscript would benefit from tempering description of “uniqueness” in sample set, particularly owing to relatively small sample size

Elsewhere: used by authors In a table and elsewhere: , “newborn screening-related conditions,”  Please note the term was used differently by the Bush Koehly:team (perhaps the original/only group using that term) ,to reflect “newborn screening-related conditions,”   that are currently or potentially identifiable through NBS programs. Hence a wider groupinWe collectively refer to these disorders as “newborn screening-related conditions,” reflecting gene and disease inclusion on the US Recommended Uniform Screening Panel Core and Secondary, NBSTRN-CR Candidates, NC-NEXUS, and others published by experts (i.e., mitochondrial disorders).” 

Due to these small, yet important considerations, this reviewer recommend "major" edits are needed. Happy to re-revire if requested to

Thank you,

Round 2

Reviewer 2 Report

Comments and Suggestions for Authors

Thank you for addressing the points made previously in my manuscript review.  I believe that inclusion of the location of the study should be included given the heterogeneity of newborn screening and long-term follow-up within the U.S. To this end, I suggest changing the title slightly: "A Qualitative Study on Families’ Experiences with Long-Term Follow-up Care in the Colorado/Wyoming Newborn Screening System."  Additionally, please modify the sentence in lines 64-66 to include the program's location: "... and engagement with LTFU as part of the Colorado/Wyoming newborn screening system."  Please modify lines 77-80 by separating it into two sentences and identifying the study hospital in the text:  "... specialty care clinics located at Children’s Hospital Colorado (Aurora, CO) [is this correct?], a large pediatric hospital. These clinics are contracted by the Colorado NBS program ...". Modify line 559 to identify the healthcare system: "...currently engaged with the Children's Hospital healthcare system..." And modify lines 571-572: "... epicenter of LTFU care in the Colorado/Wyoming newborn screening system."

Thank you for including Table 2. It would be helpful if you would further indicage the number of participants for each category, since you seem to have them, rather than the total for the group.

The paper is well written and provides information that may be helpful to other newborn screening programs.

  1.  

Reviewer 3 Report

Comments and Suggestions for Authors

Thank you to the authors for the thoughtful response to this reviewer's initial comments. Many of the comments raised have now been addressed; three remain that this reviewer considers important to clarify and very simple to mitigate.

From this reviewer's perspective, it was particularly beneficial that the authors inserted the term "engaged" in this sentence [“It should be noted that the caregivers who participated in this study are a sample that is engaged with LTFU care,..."]. This reviewer opines "engaged" can be similarly advantageous for clarity if inserted into the title and through the manuscript. 

All comments below are detailed from the perspective of this reviewer with the intent to provide useful feedback to improve this manuscript on an important topic.

While the title is somewhat clearer now,  ["A Qualitative Study on Families’ Experiences with Long-Term Follow-up Care"], this reviewer still finds the cohort in the title (and elsewhere) lacking sufficient precision and suggests adding your term, "engaged" in the title and throughout the manuscript to emphasize exclusion of those who were not engaged, as it is plausible for a family to have an experience with LTFU care and the experience is that it was never provided. This qualitative research did not include/assess that cohort.

Another point this reviewer recommends receives further consideration surrounds the term "unique." 

[for example: "While this is not the first study to gather caregivers’ input, it is unique in its breadth of perspectives (e.g., metabolic, endocrine, hemoglobinopathy, etc.) and its focus on LTFU." and 

While this is not the first study to gather families’ input on their experiences with NBS, it is unique in its focus on experiences with and perspectives on LTFU among a cohort of families that represent the spectrum of NBS disorder categories (e.g., metabolic, endocrine, hemoglobinopathy, etc.) and a wide range of children’s ages.” (page 2, lines 68-72)]

Admittedly, this reviewer has not undertaken a complete review to determine "uniqueness" as the authors stress it is. While possible, this assertion may or may not be precise, and this reviewer is always reluctant to support any claim of "uniqueness". As such, this reviewer strongly recommends adding a proviso before stating "unique"-- for example, "We believe" or "In our opinion."  To note: this reviewer recommends this for almost every paper that stresses "unique."

Lastly, the term "epicenter"  [one example: "illustrate the lived experiences of families at the epicenter of LTFU care."] lacks precision for the cohort based on a similar rationale to that explained earlier as it is plausible for a family to have an experience with LTFU care and at the "epicenter" but the experience is that it was never provided. As previously suggested, this can be clearer adding "engaged" in the title and throughout the manuscript to emphasize exclusion of those who were not engaged.

Thank you to the authors for these further considerations and contributions this research makes to the newborn screening field. 
